# Field-induced quantum spin disordered state in spin-1/2 honeycomb magnet Na$_2$Co$_2$TeO$_6$

Gaoting Lin [1], Jaehong Jeong[2,3], Chaebin Kim [2,4], Yao Wang[5,6], Qing Huang [7], Takatsugu Masuda [8,9], Shinichiro Asai[8], Shinichi Itoh [9], Gerrit Günther[10], Margarita Russina [10], Zhilun Lu [11,12], Jieming Sheng[13,14,15], Le Wang[15], Jiucai Wang [16], Guohua Wang[1], Qingyong Ren [1,13,14], Chuanying Xi[17], Wei Tong [17], Langsheng Ling[17], Zhengxin Liu [16], Liusuo Wu[15], Jiawei Mei [15], Zhe Qu [17,18], Haidong Zhou [7], Xiaoqun Wang[1], Je-Geun Park [2,4], Yuan Wan [5,6,19✉] & Jie Ma [1✉]

Spin-orbit coupled honeycomb magnets with the Kitaev interaction have received a lot of attention due to their potential of hosting exotic quantum states including quantum spin liquids. Thus far, the most studied Kitaev systems are 4d/5d-based honeycomb magnets. Recent theoretical studies predicted that 3d-based honeycomb magnets, including Na$_2$Co$_2$-TeO$_6$ (NCTO), could also be a potential Kitaev system. Here, we have used a combination of heat capacity, magnetization, electron spin resonance measurements alongside inelastic neutron scattering (INS) to study NCTO's quantum magnetism, and we have found a field-induced spin disordered state in an applied magnetic field range of 7.5 T < **B** ($\perp$ b-axis) < 10.5 T. The INS spectra were also simulated to tentatively extract the exchange interactions. As a 3d-magnet with a field-induced disordered state on an effective spin-1/2 honeycomb lattice, NCTO expands the Kitaev model to 3d compounds, promoting further interests on the spin-orbital effect in quantum magnets.

[1] Key Laboratory of Artificial Structures and Quantum Control, Shenyang National Laboratory for Materials Science, School of Physics and Astronomy, Shanghai Jiao Tong University, Shanghai 200240, China. [2] Department of Physics and Astronomy, Seoul National University, Seoul 08826, Republic of Korea. [3] Center for Correlated Electron Sciences, Institute for Basic Science (IBS), Seoul 08826, Republic of Korea. [4] Center for Quantum Materials, Seoul National University, Seoul 08826, Republic of Korea. [5] Institute of Physics, Chinese Academy of Sciences, Beijing 100190, China. [6] University of Chinese Academy of Sciences, Beijing 100049, China. [7] Department of Physics and Astronomy, University of Tennessee, Knoxville, TN 37996, USA. [8] Institute for Solid State Physics, University of Tokyo, Kashiwanoha, Kashiwa, Chiba 277-8581, Japan. [9] Institute of Materials Structure Science, High Energy Accelerator Research Organization, Tsukuba 305-0801, Japan. [10] Helmholtz-Zentrum Berlin für Materialien und Energie, Hahn-Meitner-Platz 1, Berlin 14109, Germany. [11] The Henry Royce Institute and Department of Materials Science and Engineering, The University of Sheffield, Sir Robert Hadfield Building, Sheffield S1 3JD, United Kingdom. [12] Mechanical Engineering and Design, School of Engineering and the Built Environment, Edinburgh Napier University, Edinburgh EH10 5DT, United Kingdom. [13] Spallation Neutron Source Science Center, Dongguan 523803, China. [14] Institute of High Energy Physics, Chinese Academy of Sciences, Beijing 100049, China. [15] Shenzhen Institute for Quantum Science and Engineering (SIQSE) and Department of Physics, Southern University of Science and Technology (SUSTech), Shenzhen, Guangdong 518055, China. [16] Department of Physics, Renmin University of China, Beijing 100872, China. [17] Anhui Province Key Laboratory of Condensed Matter Physics at Extreme Conditions, High Magnetic Field Laboratory, Hefei Institutes of Physical Sciences, Chinese Academy of Science, Hefei, Anhui 230031, China. [18] CAS Key Laboratory of Photovoltaic and Energy Conservation Materials, Hefei Institutes of Physical Sciences, Chinese Academy of Sciences, Hefei, Anhui 230031, China. [19] Songshan Lake Materials Laboratory, Dongguan, Guangdong 523808, China. ✉email: yuan.wan@iphy.ac.cn; jma3@sjtu.edu.cn

Magnets with significant spin–orbital couplings (SOCs) have become a new playground for quantum magnetism in recent years thanks to their potential of hosting novel quantum phases of matter[1]. A prominent example is the Kitaev model, an exactly solvable spin model that features a topological quantum spin liquid (QSL) ground state[2–5]. Microscopically, such a model could emerge from magnetic insulators with competing, spin anisotropic exchange interactions[3]. The essential prerequisites are that (a) the magnetic ions have spin–orbital entangled, Kramers degenerate ground states, and (b) they are arranged on a suitable lattice, with the two-dimensional (2D) honeycomb lattice being the simplest example[2–5]. So far, the search for the material incarnations of the Kitaev model has been focused on $4d/5d$ transition mental-based systems due to their relatively strong SOCs[4,5]. The examples include $H_3LiIr_2O_6$, $\alpha$-$Li_2IrO_3$, $\alpha$-$Na_2IrO_3$, and $\alpha$-$RuCl_3$[4–15]. $H_3LiIr_2O_6$ was initially thought to be a QSL, but later studies argue for a random singlet state resulted from the quenched disorder induced by the mobile protons[6,7]. As for $\alpha$-$Li_2IrO_3$, $\alpha$-$Na_2IrO_3$, and $\alpha$-$RuCl_3$, all of them magnetically order due to the non-Kitaev interactions[8–15]. To describe these materials, the Kitaev model has been extended to a generalized Heisenberg–Kitaev (H–K) model with five symmetry-allowed terms, Kitaev term $K$, off-diagonal symmetric exchange term $\Gamma$ and $\Gamma'$, nearest-neighbor (NN) Heisenberg coupling $J$, and the third NN Heisenberg coupling $J_3$[8–24]. $\alpha$-$RuCl_3$ is unique among these materials in that its effective Hamiltonian features a dominant Kitaev term[12,15,19–21]. Remarkably, applying an in-plane magnetic field around 7 T greatly suppresses its magnetic order and induces a potential QSL state[23,25–34].

More recently, the theoretical studies suggest that Kitaev physics could also be found in honeycomb magnets made of cobalt, a $3d$ transition metal[35–37]. In $3d^7$ Co ($t_{2g}^5 e_g^2$) compounds, the spin-active $e_g$ electrons not only generate new spin–orbit exchange channels of $t_{2g}$–$e_g$ and $e_g$–$e_g$ in addition to the $t_{2g}$–$t_{2g}$ channel operating in $d^5$ systems with $t_{2g}$-only electrons but also produce the Kitaev interaction almost entirely from the $t_{2g}$–$e_g$ process[37]. Moreover, the $t_{2g}$–$e_g$ and $e_g$–$e_g$ contributions to $\Gamma$ and $\Gamma'$ are of opposite signs and largely cancel each other[37]. This makes the cobaltates good candidates for realizing the Kitaev model[35–37]. A pertinent material is the honeycomb magnet $Na_2Co_2TeO_6$ (NCTO)[35–37]. The other example we are aware of is $BaCo_2(AsO_4)_2$, which also shows a similar field-induced disordered state at a low magnetic field[38].

As we shall present below, our studies show that NCTO fulfills all the prerequisites for Kitaev physics despite its seeming differences from the preceding $4d/5d$ compounds. Furthermore, we observe a QSL-like spin disordered state in an applied magnetic field $\mathbf{B} \perp b$-axis (parallel to Co-Co bond) and $7.5 < \mathbf{B} < 10.5$ T. This result suggests NCTO could be a novel honeycomb magnet that exhibits a field-induced disordered state and may broaden our horizon in the quest for Kitaev materials.

## Results

**Specific heat and magnetic susceptibility.** The magnetic specific heat at 0 T measured on the polycrystalline sample shows a sharp peak at $T_N = 25$ K, and its derivative shows another two anomalies at $T_F = 15$ K and $T^* = 7$ K, Fig. 1a. Their values are consistent with the previous reports[39–44]. With an increasing magnetic field, the sharp peak at $T_N$ shifts to lower temperatures and becomes broader. Eventually, it is indiscernible above 7.5 T[44], which indicates that the magnetic ordering at $T_N$ has been suppressed, and the system enters a magnetically disordered state. Supplementary Fig. 2 presents the heat capacity as a function of temperature at various fields, and the coincident heat capacity suggests no missing entropy above 50 K. The magnetic entropy $S_{mag}$ is obtained after subtracting the lattice contribution calculated by the Debye–Einstein (DE) model using Supplementary S(1)

(Supplementary Fig. 2). Integrating $C_{mag}/T$ over temperature from 100 Kelvin down to 2 Kelvin yields the residual magnetic entropy $S_{mag}(0 \text{ T}) = 3.05$ J/(mol K), which is about 27% of the theoretical total magnetic entropy, Fig. 1b. Remarkably, the residual entropy increases with increasing fields in NCTO. This unusual behavior may be contrasted with other magnets that are known to possess finite residual magnetic entropy, where the magnetic field tends to reduce the residual entropy rather than enhance it[45–48].

Another noteworthy feature is that the field dependence of the $S_{mag}$ at 17 K (the insert of Fig. 1c), reaches a maximum near 9 T. This non-monotonic behavior of $S_{mag}$ with increasing fields mirrors that of the magnetic specific heat $C_{mag}/T$ (Fig. 1a), which suggests the energy gap of the magnetic excitations closes at first and then reopens with increasing fields. It is worth mentioning that this field-dependent non-monotonic behavior was also observed in the $\alpha$-$RuCl_3$ as considered a signal of entering the filed-induced non-Abelian QSL state[25,27,29].

Figure 2 shows the DC magnetic susceptibility $\chi(T)$ measured by the zero-field cooling process on a single crystalline sample with $\mathbf{B} \perp b$-axis in the $ab$ plane. In low fields, three anomalies at $T_N$, $T_F$, and $T^*$ are observed as shown in Fig. 2a, which mirror the specific heat anomalies. With increasing fields, the peak with the antiferromagnetic (AFM) characteristics at $T_N$ shifts to lower temperatures and becomes flattened, and the other two anomalies become weaker, Fig. 2b. With $\mathbf{B} > 7.5$ T, all anomalies are indiscernible. This is consistent with the disappearance of the specific heat peak with $\mathbf{B} > 7.5$ T and suggests that the system enters a disordered phase.

The magnetization ($M$) measured at 0.5 K with $\mathbf{B} \perp b$-axis shows an anomaly around 6 T (Fig. 2c), which leads to a peak on the $dM/dB$ curve (Fig. 2d). The peak shifts to a lower field and becomes weaker with increasing temperature. Above 7.5 T, the $dM/dB$ data show two opposing temperature-dependent behaviors: the intensity of the $dM/dB$ decreases with increasing temperature below 10.5 T, whereas it increases with a temperature above 10.5 T. The $dM/dB$ curves at different temperatures all cross approximately at 10.5 T. We also note that for temperatures below 6 K and $7.5 < \mathbf{B} < 10.5$ T, when the system is in the disordered state as suggested by the specific heat and susceptibility data, the $dM/dB$ curves qualitatively coincide with each other.

We interpret our magnetization data heuristically as follows[18]. Between 7.5 and 10.5 T, the system is in a spin-disordered state with short-range spin correlations. On the one hand, the short-range spin correlations increase the $dM/dB$ intensity with increasing temperatures since these thermal fluctuations thermally activate the spins that can be flipped by the magnetic field. On the other hand, the increasing temperature thermalizes the already polarized spins to decrease the $dM/dB$ intensity. Therefore, the temperature-independent $dM/dB$ intensity below 6 K and $7.5 < \mathbf{B} < 10.5$ T indicates there exist strong short-range spin correlations for such low temperatures in the spin disordered state. Above 6 K and $7.5 < \mathbf{B} < 10.5$ T, the thermal fluctuations quickly thermalize the already polarized spins so that the $dM/dB$ intensity decreases with increasing temperature. By contrast, the system crosses over to the polarized state above 10.5 T. Thus, the temperature-independent $dM/dB$ intensity at 10.5 T reflects a characteristic field at which these two competing effects compensate each other. We, therefore, take 10.5 T as the crossover field from the correlated spin-disordered state to the spin-polarized state. The magnetization curve further suggests that the saturation field for the $\mathbf{B} \perp b$-axis case is around $B_S = 12.5$ T with saturation magnetization $M_S = 2.01\mu_B/\text{Co}^{2+}$ obtained by subtracting off the Van-Vleck paramagnetic contribution[49] (Fig. 2c).

**High-field electron spin resonance (ESR).** As shown in Fig. 3, the high-field ESR data measured at 2 K exhibits a rich excitation

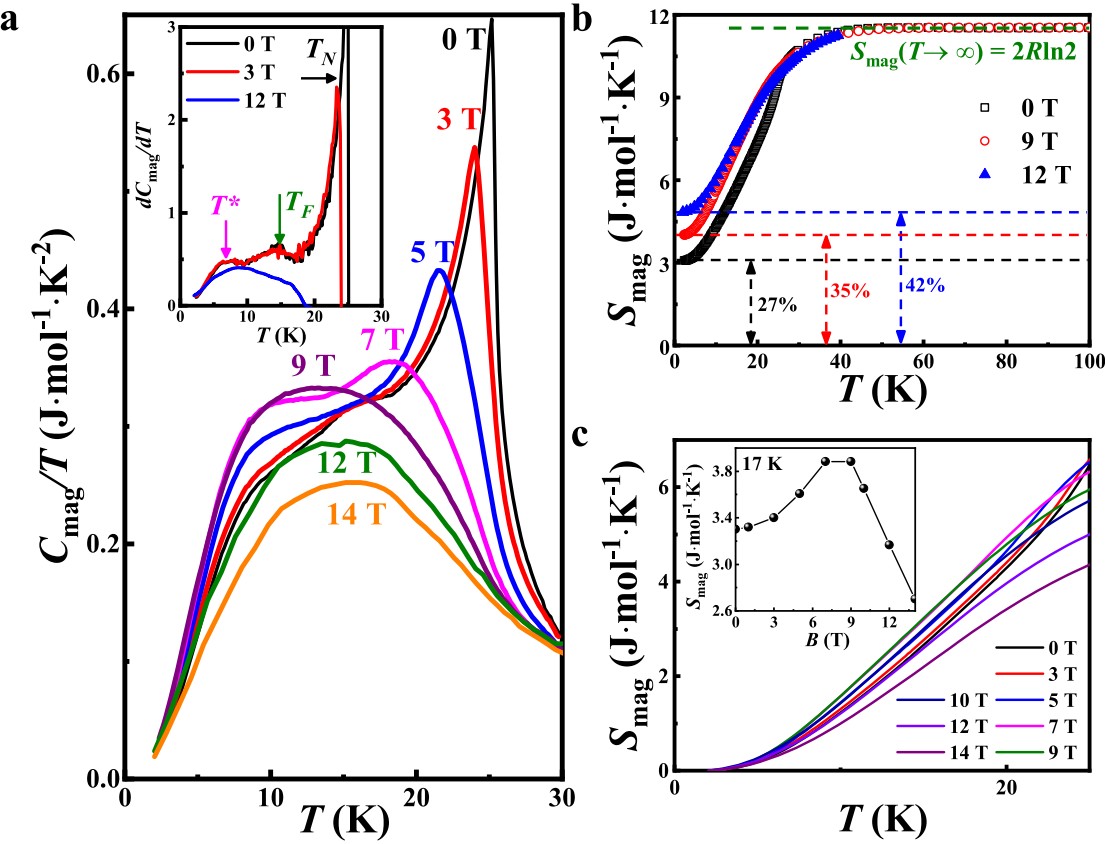

**Fig. 1 Heat capacity of Na$_2$Co$_2$TeO$_6$. a** Temperature dependence of the magnetic specific heat of NCTO for different magnetic fields up to 14 T. The inset shows the differential magnetic specific heat $dC_{mag}/dT$, and three transitions are observed as $T_N$, $T_F$, and $T^*$, respectively. **b** The magnetic entropy $S_{mag}(T)$ at 0, 9, and 12 T. The olive dashed line refers to $S_{mag}(T \rightarrow \infty)$ calculated with effective spin $J_{eff} = 1/2$ for Co$^{2+}$. In order to better observe the residual magnetic entropy, the $S_{mag}(T)$ curves are shifted vertically so that their maxima coincide with the $S_{mag}(T \rightarrow \infty)$. The black, red, and blue dashed line refer to the possible residual magnetic entropy below 2 K under different magnetic field. Compared with the total magnetic entropy $S_{mag}(T \rightarrow \infty)$, the percentage of residual magnetic entropy in different magnetic fields is shown in the figure. **c** A zoom-in of magnetic entropy below $T_N$ at various magnetic fields. Note the data are not shifted vertically as opposed to (**b**). The inset shows the field dependence of the $S_{mag}$ at 17 K.

spectrum, in which the modes A–C were observed in the ordered phase and the mode D was only detected with **B** > 6 T. The modes A–C can be described by conventional AFM resonance modes and single-magnon excitations at the Γ point[33,50], the related intensities become weaker and disappear above $T_N$ with the increasing temperature (Supplementary Fig. 3). The inset of Fig. 3k shows the electron paramagnetic resonance (EPR) spectra measured at 50 K with 214 GHz. From the resonance fields obtained by Supplementary S(2), the calculated $g$-factors are $g_{ab} = 4.13$ and $g_c = 2.3$, respectively. These values are also consistent with the magnetization data. The saturation magnetization $M_S = 2.01\mu_B$/Co$^{2+}$ for **B** ⊥ $b$-axis, Fig. 2c, leads to a $g_J = 4.02$ for the pseudospin-1/2 case. Both the ESR and the magnetization data corroborate the effective spin-1/2 model for Co$^{2+}$ ions. The higher energy crystal field levels of Co$^{2+}$ are thermally inactive in the temperature range considered in this work. The strongly anisotropic $g$ factors for Co$^{2+}$ ions in the octahedral environment confirm the strong SOC and magnetocrystalline anisotropy, which can usually drive a magnetic insulator to open a spin-wave energy gap. The extracted resonance data from Fig. 3a–j are summarized in the frequency-field diagram shown in Fig. 3k, and the extrapolation of the frequency-field dependences of modes A–C to zero field reveal a gap $\Delta E \approx 100$ GHz ≈0.4 meV.

Around 7–8 T, the AFM modes of A and B approach the EPR line with $g_{ab} = 4.13$, which again suggests a field-driven magnetically disordered state with paramagnetic-like behavior in the $ab$-plane. The field-frequency curves of the A and B modes

intersects with the EPR line near 6 T, which may be heuristically interpreted as the field at which the spin gap closes. This interpretation is also consistent with the specific heat data, which suggests the gap closes near 6 T. The field-frequency curve of the C mode gradually approaches the EPR line with $g_c = 2.3$, which indicates that its polarization is along the $c$-axis.

The ESR measurement reveals another mode that is not directly connected to the aforementioned AFM resonance modes, which we dub as D mode. It only appears when **B** > 6 T. Its excitation energy shows a linear-field dependence with a slope of ~0.15 meV/T, from which we deduce an effective $g \approx 2.6$. This effective $g$-factor is between $g_{ab}$ and $g_c$. The D mode must be associated with a magnetic excitation that only exists or becomes visible in the high-field spin disordered phase. Comparing with the other three modes, the D mode is much weaker, and its linewidth is broader. Its microscopic origin is unclear at the moment; however, we note its close resemblance with the ESR data of $\alpha$-RuCl$_3$, where new modes with linear field-energy relationship emerge in the spin disordered state[26,28,33].

**Phase diagram.** A phase diagram was constructed in Fig. 4 by combining the critical temperatures and fields obtained above. There are four regimes: (i) with **B** < 6 T, three magnetic transitions occur with decreasing temperatures. The transition at $T_N$ should be the one from paramagnetic to zigzag AFM ordering. The ones at $T_F$ and $T^*$ could be the adjustments of the canting moments of the zigzag order; (ii) For 6 T < **B** < 7.5 T, there is a

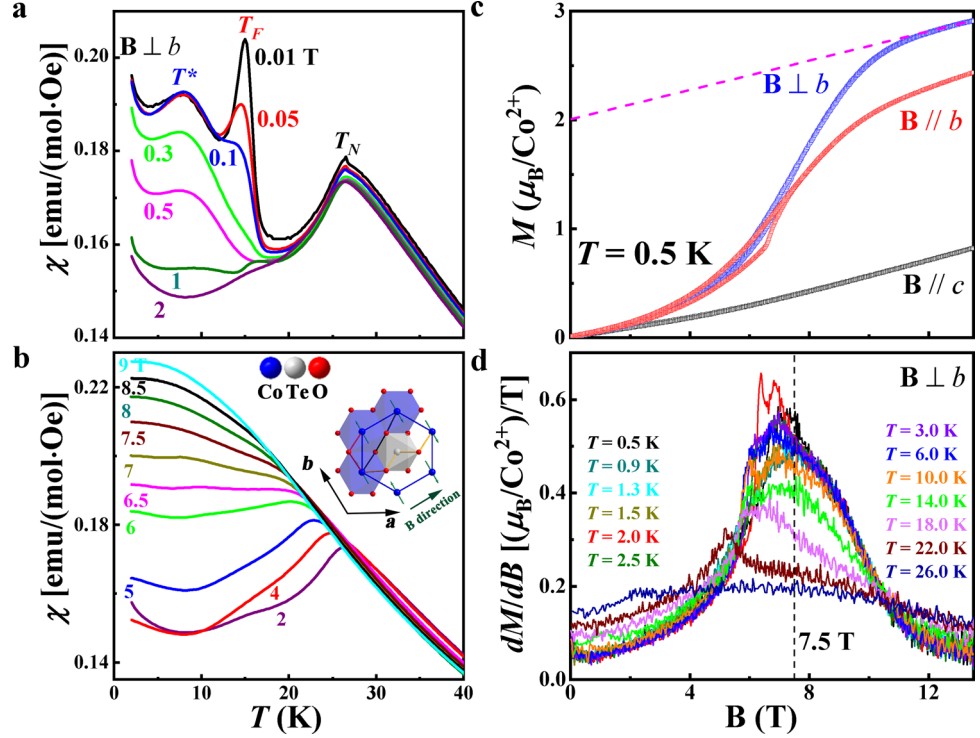

**Fig. 2 Magnetic susceptibility of Na₂Co₂TeO₆ for B ⊥ *b*-axis. a, b** Temperature dependence of susceptibility $\chi(T)$ in NCTO measured at various magnetic fields. The inset of Fig. 2b shows the honeycomb lattices of Co viewed along the *c*-axis. The cartoon shows the moments to be in the basal plane and parallel to the *b*-axis[40, 41], and the TeO₆ octahedra sit at the center of each honeycomb unit. The first NN ferromagnetic (FM) interaction *J* originated from a collaboration between Co–O–Co (red bond) superexchange interactions and Co–Co (blue bond) direct exchange interactions. The second NN $J_2$ has multiple superexchange interaction pathways, which may lead to a wide range of variations, mainly determined by *J* and $J_3$. The third NN AFM interaction $J_3$ arises from the existence of the Te atom in the center of the honeycomb lattice, which leads to the unique superexchange interaction pathway Co–O–Te–O–Co (golden bond). The olive arrow refers to the direction of applied magnetic field **B** in the magnetization. **c** The isothermal magnetization $M(B)$ with the applied magnetic field **B** ⊥ *b*, **B** // *b*, and **B** // *c*-axis at 0.5 K, respectively. The dashed line indicates the Van-Vleck paramagnetic background. **d** The differential isothermal magnetization as functions of fields $dM/dB$ vs. **B** at different temperatures with **B** ⊥ *b*.

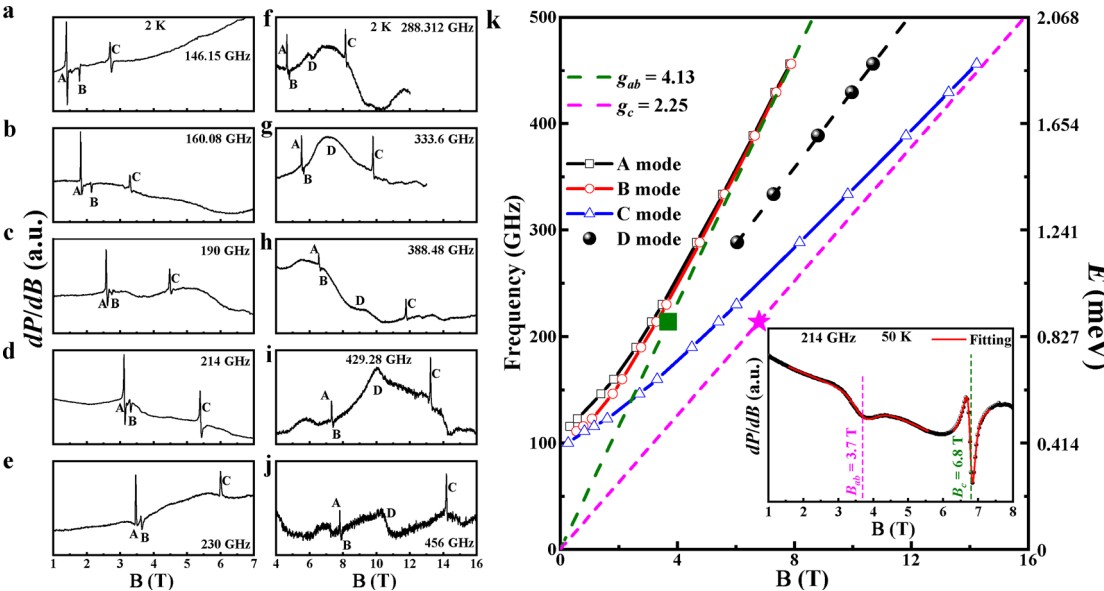

**Fig. 3 High-field ESR of Na₂Co₂TeO₆. a–j** Frequency-dependence of ESR at 2 K; **k** ESR frequency-field diagram of NCTO at 2 K. The unit conversion with meV (1 meV ≈241.8 GHz) is shown on the right axis. The inset shows the EPR spectra measured at 50 K with 214 GHz. The red lines are the fitting line by Supplementary S(2). The olive squares and magenta stars are obtained from the EPR data at 50 K.

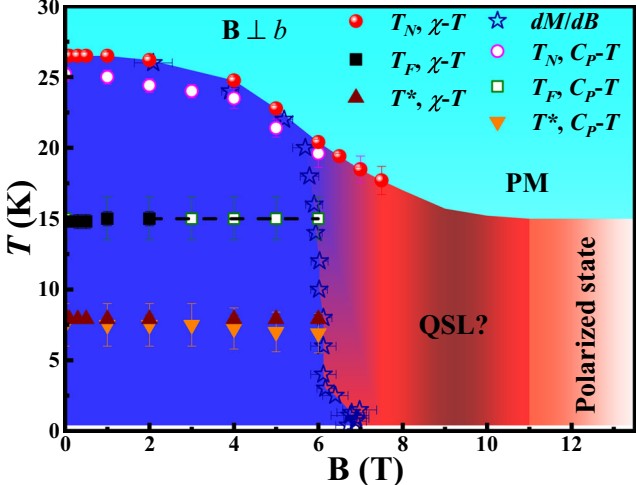

**Fig. 4 Phase diagram of Na₂Co₂TeO₆.** Temperature–magnetic field phase diagram for NCTO. The stars symbol the first-ordered phase boundary. The color background is used only as a simple guide.

coexistence of low-field magnetically ordered state and high-field spin disordered state; (iii) most interestingly, within 7.5 T < **B** < 10.5 T, the system enters a spin disordered state; (iv) with **B** > 10.5 T, the system begins to enter the polarized state and becomes fully saturated above 12.5 T.

**Spin-wave excitations.** Figure 5a presents the inelastic neutron scattering (INS) measurement with incident energy $E_i = 11.4$ meV for NCTO at 3 K and two bands (from 0.4 to 2.9 meV and from 5.9 to 7.1 meV) are observed. The magnetic mode shows an apparent minimum near $Q = 0.72$ Å$^{-1}$, which is close to the magnitude of the M point of the honeycomb reciprocal lattice as expected for a 2D magnetic system. Moreover, the concave shapes are observed at the scattering edges, similar to the magnon excitations observed in other honeycomb lattice magnets with zigzag AFM order of $\alpha$-Na₂IrO₃[51] and $\alpha$-RuCl₃[12]. As shown in Fig. 5c, d, the INS spectra show a gap of 0.4 meV at the M point. This gap is comparable in energy scale with the ESR gap, which corresponds to the excitation energy at the $\Gamma$ point.

The diffraction refinement suggested that the magnetic moments were along the crystallographic b-axis with a possible small canting toward the c-axis[40,41]. Since the inter-layer interaction is relatively weak, NCTO could be treated as a single-layer 2D compound at a first approximation. The super–super-exchange Co–O–Te–O–Co pathway produces a significant third-neighbor exchange interaction $J_3$[41]. Meanwhile, the NN Co ions can interact with each other through two 90° Co–O–Co super-exchange paths or direct AFM exchange interactions[35,41]. The cancellation between the different hopping contributions from d–d and d–p–d orbits can weaken the ferromagnetic NN J[52,53]. Together, J and $J_3$ can stabilize a zigzag magnetic order. The second neighbor $J_2$ is likely to be relatively weak in that it tends to destabilize the zigzag order. However, as $J_{1,2,3}$ is isotropic in the spin space, they cannot select the direction of the magnetic moments; this is achieved by the spin anisotropic interactions such as the K and $\Gamma$ terms mentioned in the Introduction.

The linear spin-wave theory (LSWT) is performed to analyze the INS spectra with the following exchange Hamiltonians[17–20,22,23]

$$\mathrm{H} = \sum_{\langle i,j \rangle} [J\mathbf{S}_i \cdot \mathbf{S}_j + KS_i^\gamma S_j^\gamma + \Gamma(S_i^\alpha S_j^\beta + S_i^\beta S_j^\alpha)] + J_3 \sum_{\langle\langle\langle i,j \rangle\rangle\rangle} \mathbf{S}_i \cdot \mathbf{S}_j \quad (1)$$

where, <i, j> denotes NN sites, $\mathbf{S}_i$ and $\mathbf{S}_j$ are effective spin-1/2

operators at sites i and j, respectively, $\alpha$ and $\beta$ are perpendicular to the Kitaev spin axis $\gamma$. When the $J_2$ and I (Ising exchange interaction) are close to zero (Supplementary 4.1 Symmetries and model), the zigzag AFM order will be more stable. Finally, the powder-averaged scattering numerical results are presented in Fig. 5b. With $K = \Gamma = 0.125$ meV, $J = -2.175$ meV, and $J_3 = 2.5$ meV, the calculated dispersion can reproduce qualitatively the experimental data. While the INS cannot access the excitation energy gap at the $\Gamma$ point, the LSWT calculation suggests an energy gap on the same order of magnitude as the M point gap, in qualitative agreement with the ESR results.

## Discussion

The characteristic behaviors for the field-induced spin disordered state in NCTO, such as the disappearance of the peaks observed on specific heat and $\chi(T)$, the field dependence of magnetic entropy with a maximum near 9 T below 17 K, and the existence of low energy excitations at low temperatures indicated by the $dM/dB$ curves, are all similar to those observed for the field-induced disordered state above about 7 T in $\alpha$-RuCl₃[27,29,30,54–56]. These behaviors have been believed to be evidence that $\alpha$-RuCl₃ enters a Kitaev QSL state under fields[27,29,30]. The ESR measurements of $\alpha$-RuCl₃ also show extra modes in the field-induced disordered state with a linearly increasing energy gap with increasing magnetic fields[26,33], similar to that of the D mode observed for NCTO. For $\alpha$-RuCl₃, it has been suggested that when the magnetic field and gap become large enough, it can overcome the energy scale related to the residual Heisenberg interactions so that a QSL emerges[23,26,28,32,33]. While the exact nature of this field-induced disordered state in NCTO needs further studies to determine, the similarities between the disordered states of $\alpha$-RuCl₃ and NCTO suggest the possibility of the latter being a QSL state.

We note several recent reports on the magnetic excitations of NCTO with different interpretations[57–59]. Ref. [57] performs the INS of NCTO on the same facility as this work but at the higher incident neutron beam energy of 16.54 meV. The resulted INS spectra are qualitatively similar to this work. Moreover, we obtained additional low energy spectra of NCTO with the incident energy of 3.27 meV and achieved better instrument resolution by using another neutron spectrometer NEAT II, HZB, Deutschland. The result (Fig. 5d) shows an energy gap of about 0.4 meV at the M-point. It is noticed that Ref. [57] applies a different model on the INS spectra. Ref. [57] suggests a large J (−1.5 meV) and a large AFM K (3.3 meV), whereas our simulation leads to a large J (−2.325 meV) and a small AFM K (0.125 meV). Surprisingly, both could qualitatively reproduce the INS spectra despite different choices for the relative magnitude of the K term.

Meanwhile, ref. [58] suggested a small J (−0.1 meV) and a large FM K (−9 meV); ref. [59] compared two sets of exchange interactions, which are J (−0.2 meV), K (−7 meV) and J (−3.2 meV), K (2.7 meV), respectively. In ref. [59], it was pointed out that the calculated spectra with the two models, one with FM K and the other with AFM K, are almost indistinguishable. Therefore, at this stage, it is difficult to judge which set of exchange interactions is more accurate.

The main reason for the tension among these different fitting parameters, such as a large K term vs. a small K term, or FM vs. AFM K term, could be due to the fact that all these INS measurements were performed on polycrystalline samples. The powder average effect on the weak signals makes it difficult to determine the Hamiltonian parameters for the complex systems with competing interactions and frustration. Single-crystal neutron scattering measurement in the future is critical to clarify the magnetic structure and

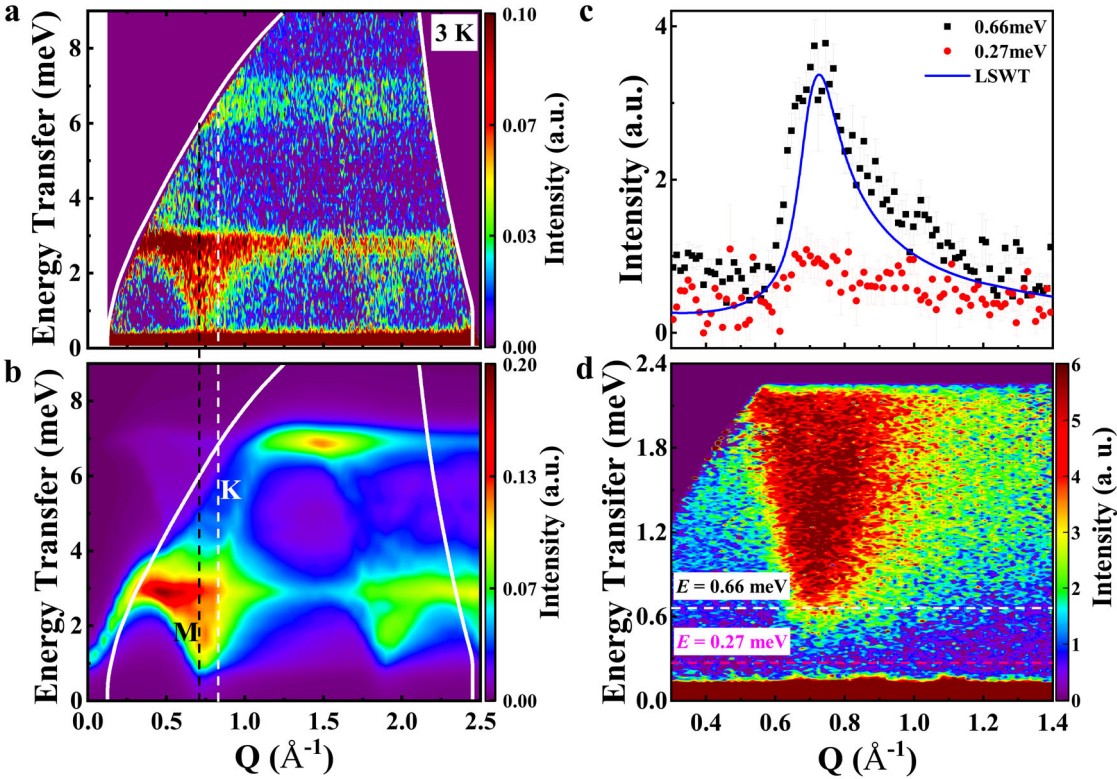

**Fig. 5 Powder INS of Na$_2$Co$_2$TeO$_6$. a** Powder INS using 11.4 meV incident neutron for NCTO at 3 K on spectrometer HRC. The high-temperature data (95 K) was subtracted as background and the thermal effect was simulated by the Bose factor. **b** The calculated powder-averaged scattering including the magnetic form factor with $K = \Gamma = 0.125$ meV, $J = -2.325$ meV, and $J_3 = 2.5$ meV. The black and white dashed lines label M and K points at the first Brillouin zone with $Q = 0.72$ and $0.83$ Å$^{-1}$, respectively. **c** Constant-$E$ cuts over the energy ranges centered at the locations labeled dashed lines in Fig. 5d. The solid blue line is constant-$E$ cuts over the energy ranges centered at 0.66 meV from our LSWT. The absence of structured scattering below 0.4 meV confirms the gap in the magnetic excitation spectrum. **d** Powder inelastic neutron scattering using 5 Å (~3.27 meV) incident neutrons for NCTO at $T = 3$ K on spectrometer NEAT II, HZB, Deutschland.

dynamics of NCTO while it could be challenging due to the small size of single crystals grown by the flux method[44].

In summary, the most significant finding from our studies on NCTO presented here is a QSL-like disordered state induced under fields with 7.5 T < **B** < 10.5 T. Therefore, NCTO is a novel example of an effective spin-1/2 honeycomb magnet that hosts a field-induced spin disordered state. Its origin, in addition to the related spin structure and spin dynamics, calls for future experimental work on single crystals and theoretical studies.

## Methods

**Sample preparation and characterization**. NCTO polycrystalline was prepared by a solid-state reaction method. At first, Na$_2$CO$_3$ (Alfa, 99.997%), Co$_3$O$_4$ (Alfa, 99.7%), and TeO$_2$ (Alfa, 99.99%) were mixed in a stoichiometric molar ratio with 5% excess Na$_2$CO$_3$, and fully ground; then, the mixture was loaded in an alumina crucible and sintered at 850 °C in the air for 40 h. The high-quality single crystal was grown by the flux method. The polycrystalline sample of NCTO was mixed with the flux of Na$_2$O and TeO$_2$ in a molar ratio of 1:0.5:2 and gradually heated to 900 °C at 3 °C/min in the air after grinding. The sample was retained at 900 °C for 30 h and was cooled to a temperature of 500 °C at the rate of 3 °C/h. The furnace was then shut down to cool to room temperature. To confirm the structure and purity of the sample, powder X-ray diffraction (XRD) measurement was performed with a HUBER imaging-plate Guinier camera 670, using Cu $K_{\alpha 1}$ radiation ($\lambda = 1.54051$ Å). The XRD patterns were refined with the Rietveld method using the conventional refinement program FullProf (Supplementary Fig. 1). The magnetic properties were checked through measurements as a function of temperature ($T$) and magnetic field (**B**) using a vibrating sample magnetometer in the physical properties measurement system (PPMS, Quantum Design). Besides, magnetization measurements were also carried out using a high-sensitive Hall sensor magnetometer[60–62] for the temperature range from 0.4 to 30 K. The heat capacity measurements were carried out using the relaxation time method in the PPMS.

**High-field ESR**. The high-field ESR measurements were performed by the high-field high-frequency electron magnetic resonance spectrometer with 25 T water-cooled resistive magnet (field sweep range: 0–25 T, frequency range: 50–690 GHz, and temperature range: 2–300 K).

**Inelastic neutron scattering**. The INS of powder NCTO was performed using the High-Resolution Chopper Spectrometer (HRC) at the J-PARC[63]. The HRC delivers high-resolution and relatively high-energy neutrons for a wide range of studies on materials dynamics. Spectra were collected at various temperatures by operating in high-flux mode (energy resolution of ≥2%$E_i$) with $E_i = 11.4$ meV. The INS of powder NCTO was also carried out on the recently upgraded cold-neutron direct-geometry time-of-flight spectrometer NEAT II, HZB, Deutschland[64,65]. Each sample was packed in aluminum cans filled with He exchange gas. Each scan was counted for around 6 h with the incident neutron energy $E_i = 3$ Å (~9 meV) and 5 Å (~3.27 meV).

## Data availability
The data that support the findings of this study are available from the corresponding authors upon request.

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

## Acknowledgements

G.T.L., Y.W., Z.Q., W.T., L.S.L, C.Y.X., and J.M. thank the financial support from the National Key Research Development Program of China (Grant nos. 2016YFA0300501,

2018YFA0704300, and 2016YFA0401802) and the National Science Foundation of China (nos. U2032213, 11774223, U1732154, 12004243, 11974396, U1832214, and 11774352). G.T.L. thanks the project funded by China Postdoctoral Science Foundation (Grant no. 2019M661474). J.M. thanks a Shanghai talent program. Y.W. thanks the Strategic Priority Research Program of the Chinese Academy of Sciences (Grant no. XDB33020300). Q.H. and H.D.Z. thank the support from NSF-DMR-2003117. Work at the CQM and SNU was supported by the Leading Researchers Program of the National Research Foundation of Korea (Grant no. 2020R1A3B2079375). A portion of this work was supported by the High Magnetic Field Laboratory of Anhui Province. The INS experiment was performed at the MLF of J-PARC under a user program (proposal No. 2019B0350).

## Author contributions

G.T.L., Y.W., H.D.Z., and J.M. conceived the project. Q.H. performed the XRD measurement and analyzed the data with help from G.T.L., Q.Y.R., and G.H.W. G.T.L, J.M.S., Q.H., and G.H.W. performed magnetization and heat capacity measurements and analyzed the data with help from J.C.W., Z.X.L, L.S.W., Z.Q., J.M., and H.D.Z. Q.H., L.W., J.W.M, and H.D.Z. synthesized the samples. J.J, C.K., T.M., S.A., S.I., G.G., M.R., Z.L., and J.M. performed neutron scattering measurements and analyzed the data with help from J.G.P., G.T.L., Q.Y.R., G.H.W., and Y.W. W.T., C.Y.X., L.S.L., and G.T.L. performed high-field ESR measurements. Y.W. and Y.W. performed LSWT calculations with help from X.Q.W. G.T.L., J.M., Y.W., and H.D.Z. wrote the paper with input from all other co-authors. All authors discussed the data and its interpretation.

## Competing interests

The authors declare no competing interests.
