## [Peer Review File · Nature Communications]

Reviewers' Comments:

Reviewer #1:

Remarks to the Author:

Report on Lin et al., "Field induced quantum spin disordered state"

This manuscript by Lin et al. presents a study of the magnetic properties of the honeycomb lattice material $\text{Na}_2\text{Co}_2\text{TeO}_6$ (abbreviated as NCTO) in the presence of an externally applied magnetic field. The principal result of interest is that a magnetic field applied in the honeycomb plane leads to a transition from an ordered magnetic state to a disordered state. This experimental observation is significant and if placed in the correct context could well be worthy of publication in Nature Communications. However, the authors have chosen to emphasize that this is significant primarily because the system, as determined by their fits to spin wave theory, has a small Kitaev interaction. As discussed below their assertion about the size of the Kitaev constant is unjustified, and in fact is contradicted by work carried out by many of the co-authors of this paper.

The experimental observation of the field induced disordered state in NCTO is a significant result for people working in the general field of "Kitaev materials". These are magnetic insulators exhibiting anisotropic competing interactions that might lead to a quantum spin liquid (QSL) ground state related to the of Kitaev's exactly solvable honeycomb lattice model. The essential ingredients are that the magnetic ions in the material have doubly degenerate ground states of mixed spin-orbital character and are arranged on a suitable lattice, with the 2D honeycomb lattice being the simplest example. The candidate octahedrally coordinated transition metal ions that have been investigated to date include those with five d electrons in a low spin state (e.g. Ir^{4+} or Ru^{3+}) or seven d electrons in a high spin state (e.g. Co^{2+}) with the Co^{2+} systems being investigated more recently. Most of these materials exhibit zigzag-related antiferromagnetic order at low temperatures, but it has been shown in the candidate materials $\alpha\text{-RuCl}_3$ and $\text{BaCo}_2(\text{AsO}_4)_2$ that relatively modest fields applied in the honeycomb plane can lead to a disordered state that might be related to the Kitaev QSL. There are many unanswered questions about the disordered states and therefore the report presented here that a similar phenomenon is present in NCTO is definitely significant to the study of Kitaev materials and perhaps QSL's in general. The susceptibility and specific heat results presented in this manuscript are the primary evidence for this and the phase diagram in figure 4 does a nice job of summarizing that.

The ESR results shown in figure 3 and the inelastic neutron scattering in figures 5d and 5e provide a measure of the magnetic energy gap, which is definitely relevant to any low temperature field-induced transitions from the ordered state. Often such a transition occurs when the field is strong enough to close the gap and this is hinted at in the discussion on page 6. A more complete discussion would be useful, including whether the gap size is consistent with the relevant energy scales where the field induced transition is observed.

The neutron scattering results presented in figures 5a and 5c appear to be very similar to those presented in the paper by Kim et al. (cond-mat arXiv:2012.06167v1) which includes INS measurement of NCTO apparently performed using the same instrument as those shown in Figures 5a and 5c, but with a slightly different incident energy. That data is certainly consistent with the data presented here. The arXiv paper is co-authored by several of the co-authors of this paper. The LSWT analysis presented in the arXiv paper, provides a good fit to the data with a totally different set of parameters presented in table 1 of that paper, and finds $K = 3.5$ meV, $J_1 = -1.2$ meV, $J_3 = 1.6$ meV, $\Gamma = -3$ meV and $\Gamma' = 3$ meV. Notably, analysis of the same data does not conclude that K is an order of magnitude smaller than J as is claimed in this paper. Moreover, Songvilay et al (PRB 102, 224429 (2020)) show similar INS data and present a similar LSWT analysis that fits the data very well, but with a totally different set of parameters that actually have a different sign for the Kitaev constant K .

This begs the question: what is one to make of these inconsistencies? There is not an obvious explanation for why several authors of this manuscript have presented inconsistent analyses without commenting on it. However, the primary issue is that for complex systems like NCTO with competing interactions and possible frustration it is very difficult to definitively determine the Hamiltonian parameters from inelastic neutron scattering data on polycrystalline material. There

are many parameter sets that might provide a reasonable fit. Given this, it is very hard to justify the claim made in this manuscript that NCTO has a very small Kitaev constant compared to other materials in the same class. Consequently, the discussion of the implications of that possibility are also not supported by the data presented here. It is strongly recommended that the authors rework the paper to highlight the experimental finding of the field-induced transition and the phase diagram and de-emphasize the speculative interpretation related to the small K constant that is advanced in the paper. They should also acknowledge and refer to the INS data in the papers mentioned above.

Further comments/questions for the authors' consideration related to the content of the paper:

- As seen in α - RuCl_3 , for fields applied in the honeycomb plane the transition fields can depend on the direction of the applied field in-plane. A comment on whether the dependence on the in-plane field direction was explored would be useful. Is this direction known for the data presented here? If so it should be stated explicitly along with the angle between the field direction in plane and a nearest neighbor Co-Co bond. (see the comment below on the presentation for one suggestion of how to do this).
- What do the authors think is special about the field of 10.5 T where the dM/dB curves cross? Is there some physics there?

Comments related to the presentation:

- The inset of figure 5d shows a cartoon of the crystal structure. It might help the readers to include such a figure closer to the beginning of the paper showing the applied field direction and defining what is meant by the a, b, and c axes. As it stands the paper seems to assume that the reader is familiar with the author's conventions for these but that will not necessarily be true. The known magnetic structure should also be defined and explained in such a figure.
- The discussion of the ESR uses both GHz and meV units – it would be useful for some readers to state the relation between them.
- The labels (a), (b), (c), (e) in figure 5 are difficult to read and a different color choice for the text might help.

Reviewer #2:

Remarks to the Author:

The paper describes a series of measurements including heat capacity, ESR, and neutron scattering on the Co^{2+} honeycomb $\text{Na}_2\text{Co}_2\text{TeO}_6$. The authors use neutron scattering to claim that K is small, while they observe a high field region that is similar to that observed in RuCl_3 . The experimental results are of high quality and the data will be of strong interest to the community, particularly given the recent active interest in Kitaev systems in the 4d/5d transition metal compounds. However, I think the conclusions based on the powder neutron inelastic data are not conclusive and therefore the claim that K is small is somewhat speculative, and also needs to be defined more clearly. Indeed, recently published work in PRB 102, 224429 (2020) comes to the opposite conclusion based on a similar analysis. I think this is only going to be resolved with single crystals. I think the authors should reconsider their statements about K and also cite the recent neutron scattering results. The report of a high field phase is very interesting, so I think the authors could reconstruct their paper softening the point about K. I am happy to reconsider the paper after the authors have done this and considered my comments.

Reviewer #3:

Remarks to the Author:

G. Lin and coworkers report the field-temperature phase diagram and spin excitations on the Co-based honeycomb material $\text{Na}_2\text{Co}_2\text{TeO}_6$. This 3d7 high-spin system has recently garnered much attention as it is hailed as a new route to achieve the celebrated Kitaev model. The authors employed several experimental techniques including thermodynamic, ESR, and INS, to figure out magnetic behaviors. The key findings are the field-induced quantum disordered state in the intermediate field range of $B=7.5 - 10.5$ T when an external magnetic field is applied along the B perp b axis. This in-field phase diagram is reminiscent of that of α - RuCl_3 . In addition,

modeling and analyzing the INS data enables the authors to determine a small yet finite Kitaev and Gamma. At the same time, however, the authors cast doubt about whether the determined spin Hamiltonian can harbor the field-induced disordered phase.

I judge that the observation of the field-induced QSL-like phase in $\text{Na}_2\text{Co}_2\text{TeO}_6$ is interesting, warranting further investigations. But the obtained magnetic parameters are subject to doubt. Although the theory can capture the energy scale of the higher-energy magnetic excitations but not their spectral weight. More seriously, the authors posted another paper in arXiv:2012.06167, claiming the leading AFM Kitaev interaction. The largely distinct spin Hamiltonian proposed by the two closely collaborating groups confuses the readers and invalidates the modeling results. Given this conflicting data analysis, I do not recommend the publication of this manuscript in Nature Communications.

In addition, the manuscript should be amended in some passages where the description of the data is either unclear.

(1) ESR: As the authors know well, the ESR data of $\alpha\text{-RuCl}_3$ show a bunch of new modes related to magnon-bound states in higher fields. In this regard, the authors are encouraged to elaborate more on the assignment of the D mode.

(2) Heat capacity in Figure S2: I think the selling point in this paper is the presence of the field-induced QSL-like phase. If this is the case, the authors are encouraged to provide more discussion on the low-temperature specific heat at $B=7.5 - 10.5$ T.

A point-by-point response to the reviewers

Reviewer #1:

This manuscript by Lin et al. presents a study of the magnetic properties of the honeycomb lattice material $\text{Na}_2\text{Co}_2\text{TeO}_6$ (abbreviated as NCTO) in the presence of an externally applied magnetic field. The principal result of interest is that a magnetic field applied in the honeycomb plane leads to a transition from an ordered magnetic state to a disordered state. This experimental observation is significant and if placed in the correct context could well be worthy of publication in Nature Communications. However, the authors have chosen to emphasize that this is significant primarily because the system, as determined by their fits to spin wave theory, has a small Kitaev interaction. As discussed below their assertion about the size of the Kitaev constant is unjustified, and in fact is contradicted by work carried out by many of the coauthors of this paper.

The experimental observation of the field induced disordered state in NCTO is a significant result for people working in the general field of “Kitaev materials”. These are magnetic insulators exhibiting anisotropic competing interactions that might lead to a quantum spin liquid (QSL) ground state related to the of Kitaev’s exactly solvable honeycomb lattice model. The essential ingredients are that the magnetic ions in the material have doubly degenerate ground states of mixed spin-orbital character and are arranged on a suitable lattice, with the 2D honeycomb lattice being the simplest example. The candidate octahedrally coordinated transition metal ions that have been investigated to date include those with five d electrons in a low spin state (e.g. Ir^{4+} or Ru^{3+}) or seven d electrons in a high spin state (e.g. Co^{2+}) with the Co^{2+} systems being investigated more recently. Most of these materials exhibit zigzag-related antiferromagnetic order at low temperatures, but it has been shown in the candidate materials $\alpha\text{-RuCl}_3$ and $\text{BaCo}_2(\text{AsO}_4)_2$ that relatively modest fields applied in the honeycomb plane can lead to a disordered state that might be related to the Kitaev QSL. There are many unanswered questions about the disordered states and therefore the report presented here that a similar phenomenon is present in NCTO is definitely significant to the study of Kitaev materials and perhaps QSL’s in general. The susceptibility and specific heat results presented in this manuscript are the primary evidence for this and the phase diagram in figure 4 does a nice job of summarizing that.

RESPONSE: We thank the reviewer for the positive comments by comparing NCTO to $\alpha\text{-RuCl}_3$ and $\text{BaCo}_2(\text{AsO}_4)_2$. By following the reviewer’s constructive suggestions, the novel field-induced phase now is more centralized and the claim about the small size of Kitaev term is softened in this revised manuscript. Moreover, two paragraphs were added at the end of the discussion section to discuss the difference among the exchange interactions reported by Refs. [57], [58], [59], and ours. Please see the details listed below.

The ESR results shown in figure 3 and the inelastic neutron scattering in figures 5d and 5e provide a measure of the magnetic energy gap, which is definitely relevant to any low temperature field-induced transitions from the ordered state. Often such a transition occurs when the field is strong enough to close the gap and this is hinted at in the discussion on page 6. A more complete discussion would be useful, including whether the gap size is consistent with the relevant energy scales where the field induced transition is observed.

RESPONSE: We thank the reviewer for raising this important question. In the revised version, we added discussions on the origin of the different ESR modes and field-dependence of the gap with the relevant energy. Specifically, we added “Around 7 ~ 8 T, the AFM modes of A and B approach the EPR line with $g_{ab} = 4.13$, which again suggests a field-driven magnetically disordered state with paramagnetic-like behavior in the ab -plane. The field-frequency curves of the A and B modes intersects with EPR line near 6 T, which may be heuristically interpreted as the field at which the spin gap closes. This interpretation is also consistent with the specific heat data, which suggests the gap closes near 6 T. The field-frequency curve of the C mode gradually approaches the EPR line with $g_c = 2.3$, which indicates that its polarization is along the c axis.”

The neutron scattering results presented in figures 5a and 5c appear to be very similar to those presented in the paper by Kim et al. (cond-mat arXiv:2012.06167v1) which includes INS measurement of NCTO apparently performed using the same instrument as those shown in Figures 5a and 5c, but with a slightly different incident energy. That data is certainly consistent with the data presented here. The arXiv paper is coauthored by several of the coauthors of this paper. The LSWT analysis presented in the arXiv paper, provides a good fit to the data with a totally different set of parameters presented in table 1 of that paper, and finds $K = 3.5$ meV, $J_1 = -1.2$ meV, $J_3 = 1.6$ meV, $\Gamma = -3$ meV and $\Gamma' = 3$ meV. Notably, analysis of the same data does not conclude that K is an order of magnitude smaller than J as is claimed in this paper. Moreover, Songvilay et al (PRB 102, 224429 (2020)) show similar INS data and present a similar LSWT analysis that fits the data very well, but with a totally different set of parameters that actually have a different sign for the Kitaev constant K .

This begs the question: what is one to make of these inconsistencies? There is not an obvious explanation for why several authors of this manuscript have presented inconsistent analyses without commenting on it. However, the primary issue is that for complex systems like NCTO with competing interactions and possible frustration it is very difficult to definitively determine the Hamiltonian parameters from inelastic neutron scattering data on polycrystalline material. There are many parameter sets that might provide a reasonable fit. Given this, it is very hard to justify the claim made in this manuscript that NCTO has a very small Kitaev constant compared to other materials in the same class. Consequently, the discussion of the implications of that

possibility are also not supported by the data presented here. It is strongly recommended that the authors rework the paper to highlight the experimental finding of the field-induced transition and the phase diagram and de-emphasize the speculative interpretation related to the small K constant that is advanced in the paper. They should also acknowledge and refer to the INS data in the papers mentioned above.

RESPONSE: We thank the reviewer for bringing up this important issue.

We sincerely apologize for the confusion that we caused between our manuscript and Ref. [58] (arXiv:2012.06167v1) without a further explanation. Actually, while we started to study NCTO, we contacted both Prof. Wan's group (Institute of Physics, CAS, China) and Prof. Park's group (Seoul National University, Korea) to seek possible theoretical support. During that time, especially during last year with COVID-19, our communication with Prof. Park's group was not efficient. Once Prof. Wan's group made progress on the INS spectra simulation, we composed this manuscript with Prof. Wan and submitted it to Nature Communications. Just after our submission, we were notified by Prof. Park's group that they have been working on the INS spectra simulation and would submit a manuscript soon (the reference [58]). Indeed, it is odd for several authors on both papers with different conclusions. However, since it is hard to justify which simulation is sufficiently accurate based on powder sample data, as the reviewers also strongly suggested, and each of the simulations potentially provides a different view of NCTO, we now resubmit the revised manuscript to Nature Communications with de-emphasizing the small K .

Moreover, although the INS data in Fig. 5(a) was measured on neutron spectrometer HRC, J-PARC, Japan as the same as one used in Ref. [58], the incident energies of neutron beam are 11.44 meV and 16.54 meV for the present manuscript and Ref. [58]. Hence, the INS spectra between the present manuscript and that of Ref. [58] are similar but different in details. Furthermore, we used neutron spectrometer NEAT II, HZB, Deutschland, with the incident energy of 3.27 meV to study the gap information of NCTO, as shown in Fig. 5(e). The measurements with a lower incident energy decide the low energy feature of spin dynamics, hence, we clearly observed an energy gap of about 0.4 meV.

We fully agree with the reviewer on that (i) the question then is which set of exchange interactions is right. In fact, besides the paper (Ref. [57] PRB 102, 224429 (2020)) mentioned by the reviewer, there is another recent paper that reported two other sets of exchange interactions by INS spectra simulation for NCTO (Ref. [59]). Therefore, there are totally five different sets of exchange interactions reported on NCTO; (ii) for complex systems like NCTO with competing interactions and possible frustration it is very difficult to definitively determine the Hamiltonian parameters from inelastic neutron scattering data on polycrystalline material; (iii) our claim of the small Kitaev term cannot be fully justified without high quality single crystal.

By taking account of all these points, in the revised manuscript, the novel field-induced phase now is more centralized and the claim about the small size of Kitaev term is softened. Moreover, two paragraphs were added at the end of the discussion section to discuss the difference among the exchange interactions reported by Refs. [57], [58], [59], and ours. Specifically, we pointed out that “Ref. [57] reported a small J (-0.1 meV) and a large FM K (-9 meV), Ref. [58] reported a large J (-1.5 meV) and a large AFM K (3.3 meV), and our simulation led to a large J (-2.325 meV) and a small AFM K (0.125 meV). In Ref. [59], two models with two sets of exchange interactions were used for comparison, which are J (-0.2 meV), K (-7 meV) and J (-3.2 meV), K (2.7 meV), respectively. Surprisingly, the simulations from all these five sets of exchange interactions can reproduce the INS spectra. In Ref. [59], it was even mentioned that the calculated spectra with the two models, one with FM K and the other with AFM K , are almost indistinguishable. Therefore, at this stage, it is difficult to judge which set of exchange interactions is more accurate.” Thereafter, we followed the reviewer’s suggestion to point out that this inconsistency could be mainly due to the fact that all these INS spectra were obtained on powder samples.

In summary, in this revised version, we relaxed the original claim on the small Kitaev term but compared the five sets of the reported exchange interactions. We keep the inconsistency among them as an open question and call for future INS data on single crystals to clarify the values of exchange interactions in NCTO.

Further comments/questions for the authors’ consideration related to the content of the paper:

As seen in α -RuCl₃, for fields applied in the honeycomb plane the transition fields can depend on the direction of the applied field in-plane. A comment on whether the dependence on the in-plane field direction was explored would be useful. Is this direction known for the data presented here? If so it should be stated explicitly along with the angle between the field direction in plane and a nearest neighbor Co-Co bond. (see the comment below on the presentation for one suggestion of how to do this).

RESPONSE: We thank the reviewer for pointing out the issue on the direction of applied field. Similar to α -RuCl₃, NCTO is antiferromagnetic at 0 T. However, the magnetic structure is more complicated at 0 T and two more magnetic transitions have been shown in Fig.4. Following the reviewer’s suggestion, we labeled the direction of the magnetic field, which is parallel to the nearest neighbor Co-Co bond or perpendicular to the b axis in ab -plane. An inset in Fig. 2(b) was added to illustrate this, which also includes the spin directions of the magnetic structure.

What do the authors think is special about the field of 10.5 T where the dM/Db curves cross? Is there some physics there?

RESPONSE: Thanks to the reviewer for raising this question. It is truly not accidental that the dM/dB curves cross at 10.5 T, which indicates that a transition occurs. At this transition point, two competing effects compensate each other: (i) the dM/dB intensity increases with increasing temperatures since the thermal fluctuations activate the spins which can be easier flipped by the magnetic field; (ii) the dM/dB intensity decreases with temperature since the increasing temperature thermalizes the already polarized spins.

Actually, as $B > 7.5$ T, the second effect dominates, partially owing to the lack of available low-energy excitations to be thermally activated. Therefore, the temperature-independence of dM/dB at 10.5 T indicates a finite density of states for low-energy excitations. Therefore, the transition from the QSL-like phase to the polarized state occurs at 10.5 T.

Comments related to the presentation:

The inset of figure 5d shows a cartoon of the crystal structure. It might help the readers to include such a figure closer to the beginning of the paper showing the applied field direction and defining what is meant by the a , b , and c axes. As it stands the paper seems to assume that the reader is familiar with the author's conventions for these but that will not necessarily be true. The known magnetic structure should also be defined and explained in such a figure.

RESPONSE: We thank the reviewer for pointing out this issue. We added an inset in Fig. 2(b) to illustrate (i) the direction of the applied magnetic field B , which is parallel to the nearest neighbor Co-Co bond, or perpendicular to the b axis in the ab plane; (ii) the spin directions of the magnetic structure.

The discussion of the ESR uses both GHz and meV units – it would be useful for some readers to state the relation between them.

RESPONSE: We thank the reviewer for the suggestion about the unit. We have labeled the meV unit on the right side of Fig. 3(k).

The labels (a), (b), (c), (e) in figure 5 are difficult to read and a different color choice for the text might help.

RESPONSE: We thank the reviewer for pointing out the color issue. We have accordingly changed the color of the labels.

Reviewer #2:

The paper describes a series of measurements including heat capacity, ESR, and

neutron scattering on the Co²⁺ honeycomb Na₂Co₂TeO₆. The authors use neutron scattering to claim that K is small, while they observe a high field region that is similar to that observed in RuCl₃. The experimental results are of high quality and the data will be of strong interest to the community, particularly given the recent active interest in Kitaev systems in the 4d/5d transition metal compounds. However, I think the conclusions based on the powder neutron inelastic data are not conclusive and therefore the claim that K is small is somewhat speculative, and also needs to be defined more clearly. Indeed, recently published work in PRB 102, 224429 (2020) comes to the opposite conclusion based on a similar analysis. I think this is only going to be resolved with single crystals. I think the authors should reconsider their statements about K and also cite the recent neutron scattering results.

RESPONSE: We thank the reviewer for the positive comments and for raising this detailed question. We agree with the reviewer that the system is too complicated to decide the dynamic parameter sets by powder neutron inelastic data. In the revised version, we have revised the title of the manuscript by removing the term “small K ”, and the section of DISCUSSION has been reconstructed by centralizing the QSL-like phase and discussing the reported sets of exchange interactions.

Specifically, by citing all other three recently reported papers studying INS spectra on NCTO, we pointed out that “ Ref. [57] reported a small J (-0.1 meV) and a large FM K (-9 meV), Ref. [58] reported a large J (-1.5 meV) and a large AFM K (3.3 meV), and our simulation led to a large J (-2.325 meV) and a small AFM K (0.125 meV). In Ref. [59], two models with two sets of exchange interactions were used for comparison, which are J (-0.2 meV), K (-7 meV) and J (-3.2 meV), K (2.7 meV), respectively. Surprisingly, the simulations from all these five sets of exchange interactions can reproduce the INS spectra. In Ref. [59], it was even mentioned that the calculated spectra with the two models, one with FM K and the other with AFM K , are almost indistinguishable. Therefore, at this stage, it is difficult to judge which set of exchange interactions is more accurate.” Thereafter, we suggest that the main reason for this inconsistency could be due to the fact that all these INS spectra were obtained on powder samples, which was also indicated by the reviewer.

In summary, in this revised version, we relaxed the claim on the small Kitaev term but compared the five sets of the reported exchange interactions. We keep the inconsistency among them as an open question and, as the reviewer pointed out, call for future INS data on single crystals to clarify the values of exchange interactions in NCTO.

The report of a high field phase is very interesting, so I think the authors could reconstruct their paper softening the point about K . I am happy to reconsider the paper after the authors have done this and considered my comments.

RESPONSE: We thank the reviewer for the positive comments and reconstructed the

manuscript by centralizing the field induced disordered phase and softening the point about the K term, as addressed above.

Reviewer #3:

G. Lin and coworkers report the field-temperature phase diagram and spin excitations on the Co-based honeycomb material $\text{Na}_2\text{Co}_2\text{TeO}_6$. This $3d^7$ high-spin system has recently garnered much attention as it is hailed as a new route to achieve the celebrated Kitaev model. The authors employed several experimental techniques including thermodynamic, ESR, and INS, to figure out magnetic behaviors. The key findings are the field-induced quantum disordered state in the intermediate field range of $B=7.5 - 10.5$ T when an external magnetic field is applied along the B perp b axis. This in-field phase diagram is reminiscent of that of $\alpha\text{-RuCl}_3$. In addition, modeling and analyzing the INS data enables the authors to determine a small yet finite Kitaev and Γ . At the same time, however, the authors cast doubt about whether the determined spin Hamiltonian can harbor the field-induced disordered phase.

I judge that the observation of the field-induced QSL-like phase in $\text{Na}_2\text{Co}_2\text{TeO}_6$ is interesting, warranting further investigations. But the obtained magnetic parameters are subject to doubt. Although the theory can capture the energy scale of the higher-energy magnetic excitations but not their spectral weight. More seriously, the authors posted another paper in arXiv:2012.06167, claiming the leading AFM Kitaev interaction. The largely distinct spin Hamiltonian proposed by the two closely collaborating groups confuses the readers and invalidates the modeling results. Given this conflicting data analysis, I do not recommend the publication of this manuscript in Nature Communications.

RESPONSE: We thank the reviewer for the positive comments on the field induced QSL-like phase and bringing up this important issue on INS spectra analysis.

The issue is the same as that pointed out by the reviewer #1, and we reproduce our reply below:

We sincerely apologize for the confusion that we caused between our manuscript and Ref. [58] (arXiv:2012.06167v1) without a further explanation. Actually, while we started to study NCTO, we contacted both Prof. Wan's group (Institute of Physics, CAS, China) and Prof. Park's group (Seoul National University, Korea) to seek possible theoretical support. During that time, especially during last year with COVID-19, our communication with Prof. Park's group was not efficient. Once Prof. Wan's group made progress on the INS spectra simulation, we composed this manuscript with Prof. Wan and submitted it to Nature Communications. Just after our submission, we were notified by Prof. Park's group that they have been working on the INS spectra simulation and would submit a manuscript soon (the reference [58]).

Indeed, it is odd for several authors on both papers with different conclusions. However, since it is hard to justify which simulation is sufficiently accurate based on powder sample data, as the reviewers also strongly suggested, and each of the simulations potentially provides a different view of NCTO, we now resubmit the revised manuscript to Nature Communications with de-emphasizing the small K .

Moreover, although the INS data in Fig. 5(a) was measured on neutron spectrometer HRC, J-PARC, Japan as the same as one used in Ref. [58], the incident energies of neutron beam are 11.44 meV and 16.54 meV for the present manuscript and Ref. [58]. Hence, the INS spectra between the present manuscript and that of Ref. [58] are similar but different in details. Furthermore, we used neutron spectrometer NEAT II, HZB, Deutschland, with the incident energy of 3.27 meV to study the gap information of NCTO, as shown in Fig. 5(e). The measurements with a lower incident energy decide the low energy feature of spin dynamics, hence, we clearly observed an energy gap of about 0.4 meV.

We fully agree with the reviewer on that (i) the question then is which set of exchange interactions is right. In fact, besides the paper (Ref. [57] PRB 102, 224429 (2020)) mentioned by the reviewer, there is another recent paper that reported two other sets of exchange interactions by INS spectra simulation for NCTO (Ref. [59]). Therefore, there are totally five different sets of exchange interactions reported on NCTO; (ii) for complex systems like NCTO with competing interactions and possible frustration it is very difficult to definitively determine the Hamiltonian parameters from inelastic neutron scattering data on polycrystalline material; (iii) our claim of the small Kitaev term cannot be fully justified without high quality single crystal.

By taking account of all these points, in the revised manuscript, the novel field-induced phase now is more centralized and the claim about the small size of Kitaev term is softened. Moreover, two paragraphs were added at the end of the discussion section to discuss the difference among the exchange interactions reported by Refs. [57], [58], [59], and ours. Specifically, we pointed out that “Ref. [57] reported a small J (-0.1 meV) and a large FM K (-9 meV), Ref. [58] reported a large J (-1.5 meV) and a large AFM K (3.3 meV), and our simulation led to a large J (-2.325 meV) and a small AFM K (0.125 meV). In Ref. [59], two models with two sets of exchange interactions were used for comparison, which are J (-0.2 meV), K (-7 meV) and J (-3.2 meV), K (2.7 meV), respectively. Surprisingly, the simulations from all these five sets of exchange interactions can reproduce the INS spectra. In Ref. [59], it was even mentioned that the calculated spectra with the two models, one with FM K and the other with AFM K , are almost indistinguishable. Therefore, at this stage, it is difficult to judge which set of exchange interactions is more accurate.” Thereafter, we followed the reviewer’s suggestion to point out that this inconsistency could be mainly due to the fact that all these INS spectra were obtained on powder samples.

In summary, in this revised version, we relaxed the original claim on the small Kitaev

term but compared the five sets of the reported exchange interactions. We keep the inconsistency among them as an open question and call for future INS data on single crystals to clarify the values of exchange interactions in NCTO.

In addition, the manuscript should be amended in some passages where the description of the data is either unclear.

(1) ESR: As the authors know well, the ESR data of α -RuCl₃ show a bunch of new modes related to magnon-bound states in higher fields. In this regard, the authors are encouraged to elaborate more on the assignment of the D mode.

RESPONSE: Many thanks to the reviewer for this important suggestion. In the revised manuscript we added a paragraph to specifically discuss the new mode (the D mode) in higher fields. In details, we added “The ESR measurement reveals another mode that is not directly connected to the aforementioned AFM resonance modes, which we dub as D mode. It only appears when $B > 6$ T. Its excitation energy shows a linear-field dependence with a slope of ~ 0.15 meV/T, from which we deduce an effective $g \approx 2.6$. This effective g -factor is between g_{ab} and g_c . The D mode must be associated with a magnetic excitation that only exists or becomes visible in the high-field spin disordered phase. Compared with the other three modes, the D mode is much weaker, and its linewidth is broader. Its microscopic origin is unclear at the moment; however, we note its close resemblance with the ESR data of α -RuCl₃, where new modes with linear field-energy relationship emerge in the spin disordered state”.

(2) Heat capacity in Figure S2: I think the selling point in this paper is the presence of the field-induced QSL-like phase. If this is the case, the authors are encouraged to provide more discussion on the low-temperature specific heat at $B=7.5 - 10.5$ T.

RESPONSE: We thank the reviewer for this constructive comment and suggestion. (i) We have revised Fig.1(b) to exhibit the magnetic entropy change under applied fields in more details. We added discussion to point out that this entropy change under fields is unusual since the residual entropy increases with increasing fields in NCTO while for common magnets the magnetic field tends to reduce the residual entropy rather than enhance it; (ii) We further added a sentence to emphasize this field-dependent non-monotonic behavior of magnetic entropy was also observed in the α -RuCl₃ and is believed to be a signal of entering the field-induced non-Abelian QSL state. We hope that these added discussions further strengthen properly the presentation of a QSL-like phase based on the low temperature specific heat.

Reviewers' Comments:

Reviewer #1:

Remarks to the Author:

Second review of NCOMMS-20-47594A

The authors have significantly reworked the paper and change the emphasis to better reflect the significance of the new data presented. As stated in the first report the observation in NCTO of a field-induced transition to a magnetically disordered state is sufficiently interesting to merit publication in Nature Communications.

Some comments for the authors' consideration:

- As discussed in the literature, the specific heat of a Kitaev QSL is expected to show two peaks as a function of temperature. This structure seems to have been seen in RuCl₃ after correcting for the lattice specific heat. It would be useful to comment on whether the data rules this out in NCTO or whether a second peak might exist at higher T, and if so how this compares to that observed in RuCl₃.
- Why are the asymptotic values at T=0 different in figures 1 (b) and (c)? Were the residual values subtracted off? I didn't see an explanation in the text and found this a bit confusing.
- It is stated in the section on ESR that the saturation magnetization was determined after subtracting the paramagnetic Van Vleck contribution. It would be helpful to the reader if this statement was moved up to the discussion on the susceptibility and magnetization.
- It should be noted that the J=1/2 model for the Co²⁺ ground state is only valid at low T and depending on the local environment the lowest manifold may have 6 Kramers doublets. These will split in a magnetic field and at high T some of them will be thermally populated. Could this have any effect on the field and temperature dependence? The answer is very probably no but a short comment may be useful.
- (No action required by the authors on this) – the expression used for the powder average in the supplementary material is an integral over all orientations of the wavevector. This calculation of a powder average implicitly assumes that the instrumental resolution is perfect in energy and wave vector. With imperfect resolution one has to consider all orientations of the crystal, including rotations about the wavevector. Given the assumptions made in the present manuscript this is not significant but it can matter under some circumstances.

Reviewer #2:

Remarks to the Author:

The paper describes the magnetic field and temperature phase diagram of a candidate Kitaev material based on Cobalt. The main focus in this field has been to use 4d/5d transition elements with large spin-orbit coupling to search for such materials. The paper now emphasizes the high field phase diagram and the comparison with α -RuCl₃. The paper combines magnetization, heat capacity, and ESR to establish this phase diagram, I thank the authors for considering my previous comments and changing the manuscript. I think the paper has been improved considerably and should be published.

One minor point that I would like to suggest is that I think it would help to link the ESR and neutron parts more closely. The authors claim that the ESR data is measuring magnetic excitations near the Γ point and then use the field dependence to establish the phase diagram. The authors have neutron results (though powder averaged) at this point and I think it would be helpful to compare the results qualitatively for consistency.

Overall, I think the papers has improved considerably over the previous submission and I recommend that it be published.

Reviewer #3:

Remarks to the Author:

G. Lin and co-workers have addressed most of the original concerns. In particular, the authors provide a fair description of the reported INS analysis, emphasizing the field-induced putative QSL.

Although a detailed picture is still lacking, I am in favour of publication in Nature Communications.

A point-by-point response to the reviewers

Reviewer #1:

Second review of NCOMMS-20-47594A

The authors have significantly reworked the paper and change the emphasis to better reflect the significance of the new data presented. As stated in the first report the observation in NCTO of a field-induced transition to a magnetically disordered state is sufficiently interesting to merit publication in Nature Communications.

Some comments for the authors' consideration:

As discussed in the literature, the specific heat of a Kitaev QSL is expected to show two peaks as a function of temperature. This structure seems to have been seen in RuCl₃ after correcting for the lattice specific heat. It would be useful to comment on whether the data rules this out in NCTO or whether a second peak might exist at higher T , and if so how this compares to that observed in RuCl₃.

RESPONSE: For our specific heat data, no obvious second peak was observed at high temperatures after subtracting off the phonon contribution by using the Debye-Einstein model. However, we certainly cannot rule out the possibility that such a feature exists in NCTO, since the subtraction process using Debye-Einstein model may not be perfect at high temperatures. As this work is dedicated to the low temperature properties of NCTO and the field-induced phases, the question about whether there is another high temperature specific heat peak in NCTO is beyond our scope. In the future, we plan to measure the specific heat on single crystal NCTO samples and synthesize non-magnetic analogue of NCTO to achieve better phonon background and therefore further clarify this issue.

Why are the asymptotic values at $T=0$ different in figures 1 (b) and (c)? Were the residual values subtracted off? I didn't see an explanation in the text and found this a bit confusing.

RESPONSE: In Fig. 1(b), we shifted the entropy curves vertically so that the entropy saturates at $2R\ln(2)$ in the high temperature limit. In Fig.1(c), we did not shift the entropy curves.

In the revised manuscript, we have added a couple of sentences to the caption of Fig.1 to explain the difference between these two panels.

It is stated in the section on ESR that the saturation magnetization was determined after subtracting the paramagnetic Van Vleck contribution. It would be helpful to the reader if this statement was moved up to the discussion on the susceptibility and magnetization.

RESPONSE: We have moved this sentence to the end of the "Specific heat and

magnetic susceptibility” section following the referee’s suggestion.

It should be noted that the $J=1/2$ model for the Co^{2+} ground state is only valid at low T and depending on the local environment the lowest manifold may have 6 Kramers doublets. These will split in a magnetic field and at high T some of them will be thermally populated. Could this have any effect on the field and temperature dependence? The answer is very probably no but a short comment may be useful.

RESPONSE: Indeed, the spin-1/2 model is an effective, low-energy description for the Co^{2+} ions. There are also high energy crystal field levels above the ground state doublet. However, these levels are too high in energy to be active in the temperature/field range considered in this work. Indeed, both the thermodynamic measurement and the ESR measurement are fully consistent with the spin-1/2 model.

In the revised manuscript, we have added a brief discussion on the effective spin-1/2 model and the high energy crystal field levels in the “High-field electron spin resonance” section.

(No action required by the authors on this) – the expression used for the powder average in the supplementary material is an integral over all orientations of the wavevector. This calculation of a powder average implicitly assumes that the instrumental resolution is perfect in energy and wave vector. With imperfect resolution one has to consider all orientations of the crystal, including rotations about the wavevector. Given the assumptions made in the present manuscript this is not significant but it can matter under some circumstances.

RESPONSE: We thank the reviewer for pointing out this issue. We agree that a proper treatment must include a momentum-energy dependent instrument function, which we substitute by a simple broadening procedure in this work. As the reviewer has pointed out, incorporating the exact instrument function is unlikely to significantly change the over all features of the calculated spectra. Since the comparison between the linear spin wave theory calculation and the measured spectra is qualitative, we think it is sufficient for our purpose to use a simple broadening procedure.

Reviewer #2:

The paper describes the magnetic field and temperature phase diagram of a candidate Kitaev material based on Cobalt. The main focus in this field has been to use 4d/5d transition elements with large spin-orbit coupling to search for such materials. The paper now emphasizes the high field phase diagram and the comparison with $\alpha\text{-RuCl}_3$. The paper combines magnetization, heat capacity, and ESR to establish this phase diagram, I thank the authors for considering my previous comments and changing the manuscript. I think the paper has been improved considerably and should be published.

One minor point that I would like to suggest is that I think it would help to link the ESR and neutron parts more closely. The authors claim that the ESR data is measuring magnetic excitations near the Γ point and then use the field dependence to establish the phase diagram. The authors have neutron results (though powder averaged) at this point and I think it would be helpful to compare the results qualitatively for consistency.

Overall, I think the papers has improved considerably over the previous submission and I recommend that it be published.

RESPONSE: The ESR measurement was performed in magnetic field and extrapolated to the zero field limit. To fully correlate the ESR data and the inelastic neutron scattering data, we would have to perform the inelastic neutron scattering experiment in magnetic field, which is beyond the scope of this work. Nonetheless, the zero-field spin excitation gap extrapolated from the ESR data and the gap measured by the neutron scattering are comparable in energy scales. Although it is difficult to fully fit these, we note the linear spin wave theory shares the same feature, namely the energy scale of the two excitation gaps are comparable.

In the revised manuscript, we have added a brief discussion on these two gaps in the “Spin-wave excitation” section.